# Blow-Up Dynamics and Synchronization in Tri-Trophic Food Chain Models

Eric M. Takyi [1], Rana D. Parshad [2,*], Ranjit Kumar Upadhyay [3] and Vikas Rai [4]

[1]  Department of Mathematics and Computer Science, Ursinus College, Collegeville, PA 19426, USA
[2]  Department of Mathematics, Iowa State University, Ames, IA 50011, USA
[3]  Department of Mathematics and Computing, Indian Institute of Technology (ISM), Dhanbad 826004, India
[4]  School of Computational and Integrative Sciences, Jawaharlal Nehru University, New Delhi 110067, India
[*]  Correspondence: rparshad@iastate.edu

**Abstract:** Synchronization of food chain models is an intensely investigated area in dynamical systems. Two pioneering models in three species food chain systems exhibiting chaotic dynamics are the Hastings and Powell (HP) model and the Upadhyay and Rai (UR) model. These are known to synchronize, even though the top predators in the two models behave differently. In the current manuscript, we show that although the HP and UR models synchronize for certain initial conditions, they *do not* synchronize for arbitrarily large initial conditions due to the blow-up dynamics present in the UR model. Thus, the synchronization of these model systems is purely a local (in initial data) phenomenon. Interestingly, we find that a similar result holds for the modified UR model as well, which has global in-time solutions for any positive initial condition. Thus, the lack of synchrony could also be attributed to the difference in the top predator's feeding preferences in the model systems. Our results have large-scale applications to population synchrony in tri-trophic food chains.

**Keywords:** synchronization; ordinary differential equations (ODEs); finite-time blow-up; food chain model





## 1. Introduction

Synchronization is a multi-disciplinary phenomenon with a wide range of applications in chemical, physical, and biological systems [1,2]. Synchronization is pivotal to our understanding of some natural systems, such as natural rhythms of clapping [3], cardiac pacemakers [4], neuronal processing [5], and cell cycles [6]. Meanwhile, the excess of synchronization in some systems may result in adverse effects such as epilepsy [7], Parkinson's disease [8], and bridge oscillations [9]. Synchronization is the agreement of different processes in time. It is a technique used in achieving related dynamics between chaotic systems by coupling. The related dynamics are asymptotic with time. The trajectories of the response system with that of the driver system are tracked. In 1990, Pecora and Carroll showed that two chaotic systems were able to synchronize with each other [10]. Since then, a wide variety of methods have been devised for the synchronization of chaotic systems. These include generalized synchronization (GS), complete synchronization (CS), and phase synchronization (PS), among others [11]. Recently, numerical and experimental studies on multistability and synchronization in heterogeneous interconnected networks were carried out in [12]. The synchronous behaviors of coupled biological neurons and four-dimensional electronic neurons, known to produce chaotic spiking–bursting behavior, were been observed in [13].

In this paper, we focus our attention on GS. When two coupled systems are under investigation, GS arises when the state or behavior of one of the coupled systems is governed or influenced by the behavior of the other system. The two systems may not necessarily be identical. GS is often seen in systems with unidirectional coupling [14]. Roy et al. [15] used a reversed engineering technique with the objective of obtaining a

GS state as a functional relation between a driver and a response oscillator. They were able to design an appropriate coupling between the driver and the response using an open-plus-closed-loop (OPCL) coupling and guaranteed the stability of the GS state. Their results showed that the OPCL method was robust to mismatching in oscillators and also viable for nonidentical oscillators. The OPCL method is applicable to any model-based system where $\dot{x} = F(\mathbf{x}, t)$, $\mathbf{x} \in \mathbb{R}^n$, which can be driven [16] and particularly chaos control related [17]. Other techniques of exploring GS in coupled systems such as the mutual false nearest neighbor were thoroughly discussed in [18]. Abarbanel et al. [19] also studied GS in drive-response systems using the auxiliary system approach. The method uses a second identical response in monitoring synchronized motions. It can also detect and characterize forced GS.

Chaotic dynamics have been observed in some food chain dynamical systems. Hastings and Powell derived a three-species food chain system that exhibited chaotic dynamics [20]. Shortly thereafter, Upadhyay and Rai derived yet another three-species food chain model exhibiting chaos. These models have been extremely thoroughly investigated, as they exhibit a variety of rich dynamics ranging from extinction to periodic dynamics to chaos and, most recently, finite-time blow-ups [21–26]. Species extinction and loss of biodiversity are two coupled issues which have become increasingly relevant in ecology. Thus, in an attempt to gain insight into an ecosystem, where there may be interactions among different food chains, Upadhyay and Rai [11] explored synchronization of the population fluctuations of their model with those of Hastings and Powell [20]. One of the key differences between the two models is that the former's top predator is a generalist, and the latter's predator at the same tri-trophic level is a specialist. A generalist predator has several food choices. Of course, it may have a food item (prey) of its liking. On the other hand, a specialist predator dies out exponentially due to a lack of additional food choices.

The UR model exhibits chaotic dynamics in a narrow parameter range (i.e., discontinuous parameter intervals). On the other hand, the HP model displays chaos in a broad, continuous parameter range. Synchronization of the chaotic dynamics of two drastically different dynamical systems has interesting ecological implications. Synchrony among two chaotic ecological systems may seem to be a mathematical artifact, but it is also evidence of an ecological principle that generalist predators are dominant contributors that *balance* food chains and webs in the wild. In [11], Upadhyay and Rai showed that the UR and HP models can indeed synchronize, implying that under certain coupling mechanisms, two drastically different food chains could synchronize. Therefore, we undertake a further exploration of the synchronization between these model systems. In this work, we use the OPCL coupling method for GS in the UR and HP models. We show the following:

- The UR and HP models individually can exhibit chaotic dynamics for the same parameter regimes. However, they will synchronize when coupled accordingly.
- This synchronization will occur only for small-to-moderate initial conditions.
- For larger initial conditions, the UR and HP models will *not* synchronize. This is shown numerically and analytically.
- For small initial conditions, the modified UR and HP models will synchronize, but for larger initial conditions, it is numerically seen that they will *not* synchronize.
- Thus, we reaffirm that the synchronization of three species' food chains with different top-down control (differently behaving top predators) is caused solely by the top predator.

## 2. Generalized Synchronization Using the OPCL Coupling Method

**Definition 1.** *Let $\mathbf{w} \in \mathbb{R}^n$. Then, we define $||\mathbf{w}||_\infty = \sup\{|w_i| : i = 1, 2, \ldots, n\}$.*

**Definition 2.** *Given a differentiable functional relation $\phi : \mathbb{R}^n \to \mathbb{R}^n$ between the state variables of the driver and response system, if there exists a controller $D(y(t), g(t))$ such that $||y(t) - \phi(g(t))||_\infty \to 0$ as $t \to \infty$, then the coupled system exhibits the property of generalized synchronization [27].*

We define the OPCL coupling term (also the controller) as

$$D(y(t), g(t)) = \dot{g}(t) - G(g(t)) + (H - J(g(t)))(y(t) - g(t)), \tag{1}$$

where $g(t) = \alpha x(t)$ is the goal state and $\alpha$ is a multiplicative factor [28]. Here, $\alpha$ can be referred to as an $n \times n$ transformation matrix whose elements may be constants, state variables of the driver system, or a combination of both. In general, we define the goal state of a three-dimensional system as

$$\begin{pmatrix} g_1 \\ g_2 \\ g_3 \end{pmatrix} = \begin{pmatrix} \alpha_{11} & \alpha_{12} & \alpha_{13} \\ \alpha_{21} & \alpha_{22} & \alpha_{23} \\ \alpha_{31} & \alpha_{32} & \alpha_{33} \end{pmatrix} \begin{pmatrix} x_1 \\ x_2 \\ x_3 \end{pmatrix}. \tag{2}$$

In addition, $H$ is an $N \times N$ constant Hurwitz matrix with all its eigenvalues having negative real parts.

**Remark 1.** $J(g(t))$ *is the Jacobian of* $G(g(t))$.

The OPCL coupling term is added to the RHS of the response system to obtain the desired response dynamics. We write the GS scheme as

$$\begin{cases} \dot{x}(t) = F(x(t)), \\ \dot{y}(t) = G(y(t)) + D(y(t), g(t)). \end{cases} \tag{3}$$

We define the error function of the coupled system as $e(t) = y(t) - g(t)$ and write $G(y)$ using Taylor series expansion as

$$G(y) = G(g) + \frac{\partial G(g)}{\partial g}(y - g) + \dots \tag{4}$$

We keep the first-order terms of the series expansion and substitute them into Equation (3) to obtain the error dynamics $\dot{e} = He$. Since $H$ is a Hurwitz matrix, then $e \to 0$ as $t \to \infty$ and thus obtain an asymptotically stable GS. The reader is referred to [15] for how to construct the $H$ matrix.

## 3. Model Systems

### 3.1. Upadhyay–Rai (UR) Model

This is a continuous-time three-species model known to exhibit chaotic dynamics [29]. It describes an ecological community where a predator population $x_2$ predates a prey population $x_1$, and $x_2$ is a specialist predator. The population $x_2$ also becomes the favorite food for individuals belonging to population $x_3$. The UR model is given by

$$\begin{aligned} \dot{x}_1 &= a_1 x_1 - b_1 x_1^2 - \frac{w x_1 x_2}{x_1 + D}, \\ \dot{x}_2 &= -a_2 x_2 + \frac{w_1 x_1 x_2}{x_1 + D_1} - \frac{w_2 x_2 x_3}{x_2 + D_2}, \\ \dot{x}_3 &= c x_3^2 - \frac{w_3 x_3^2}{x_2 + D_3}. \end{aligned} \tag{5}$$

Please see Table 1 for parameter descriptions for the UR model.

**Table 1.** Model symbols and description for Upadhyay and Rai model.

| Symbols | Description |
| --- | --- |
| $x_1$ | prey |
| $x_2$ | middle predator |
| $x_3$ | top predator |

**Table 1.** *Cont.*

| Symbols | Description |
|---|---|
| $a_1$ | intrinsic growth rate of prey |
| $b_1$ | measure of competition among prey |
| $a_2$ | intrinsic death rate of $x_2$ in the absence of food $x_1$ only |
| $D, D_1$ | measure of the level of protection offered to the prey by the environment |
| $D_2$ | value of $x_2$ at which its per capita removal rate becomes $w_2/2$ |
| $D_3$ | Loss of $x_3$ due to lack of favorite food $x_2$ |
| $c$ | growth rate of $x_3$ via sexual reproduction |
| $w, w_i's$ | maximum value that per capital rate can attain |

*3.2. Hastings–Powell (HP) Model*

Hastings and Powell in 1991 [20] showed the occurrence of chaos in a tri-trophic food chain model. A Holling type II functional response was used for both predator populations ($y_2$ and $y_3$). The model is given as

$$\dot{y_1} = ly_1 - my_1^2 - \frac{wy_1y_2}{y_1 + D},$$

$$\dot{y_2} = -a_2y_2 + \frac{w_1y_1y_2}{y_1 + D_1} - \frac{w'y_2y_3}{y_2 + D_2}, \tag{6}$$

$$\dot{y_3} = -ny_3 + \frac{w_3y_2y_3}{y_2 + D_3}.$$

A detailed description of the parameters in the system in Equation (6) can be found in [20].

## 4. Generalized Synchronization of the UR Model and HP Model Using the OPCL Coupling Method

In this section, we use the OPCL coupling technique to synchronize the UR model in Equation (5) and the HP model in Equation (6). We considered the system in Equation (5) to be the driver system $\dot{x}(t) = F(x(t))$, $x(t) \in \mathbb{R}^n$ and the system in Equation (6) to be the response system $\dot{y}(t) = G(y(t))$, $y(t) \in \mathbb{R}^n$. We compute the Jacobian of the system in Equation (6) and obtain

$$J(y) = \begin{pmatrix} l - 2my_1 - \dfrac{Dwy_2}{(D + y_1)^2} & \dfrac{-wy_1}{y_1 + D} & 0 \\ \dfrac{D_1w_1y_2}{(D_1 + y_1)^2} & -a_2 + \dfrac{w_1y_1}{y_1 + D_1} - \dfrac{D_2w'y_3}{(y_2 + D_2)^2} & -\dfrac{w'y_2}{y_2 + D_2} \\ 0 & \dfrac{w_3D_3y_3}{(y_2 + D_3)^2} & -n + \dfrac{w_3y_2}{y_2 + D_3} \end{pmatrix}. \tag{7}$$

From the Jacobian, we can write the constant H-matrix as

$$H = \begin{pmatrix} h_1 & h_2 & 0 \\ h_3 & h_4 & h_5 \\ 0 & h_6 & h_7 \end{pmatrix}. \tag{8}$$

We required the constant H-matrix to satisfy the Routh–Hurwitz criterion and thus chose the parameters $h_1 = -2, h_2 = 0, h_3 = 0, h_4 = -1, h_5 = 0, h_6 = 0$, and $h_7 = -3$. We next chose a transformation matrix $\alpha$ with arbitrary constant elements to enable us to achieve a desired goal dynamic. For example, let

$$\alpha = \begin{pmatrix} -2 & 0 & 1 \\ -2 & 0 & 0 \\ 2 & -1 & 1 \end{pmatrix}. \tag{9}$$

We achieved the goal dynamics

$$
\begin{pmatrix} g_1 \\ g_2 \\ g_3 \end{pmatrix} = \begin{pmatrix} -2 & 0 & 1 \\ -2 & 0 & 0 \\ 2 & -1 & 1 \end{pmatrix} \begin{pmatrix} x_1 \\ x_2 \\ x_3 \end{pmatrix} = \begin{pmatrix} -2x_1 + x_3 \\ -2x_1 \\ 2x_1 - x_2 + x_3 \end{pmatrix},
\tag{10}
$$

In addition, the coupling term is given by

$$
D = \begin{pmatrix} -2\dot{x}_1 + \dot{x}_3 \\ -2\dot{x}_1 \\ 2\dot{x}_1 - \dot{x}_2 + \dot{x}_3 \end{pmatrix} - \begin{pmatrix} lg_1 - mg_1^2 - \dfrac{wg_1g_2}{g_1 + D} \\ -a_2g_2 + \dfrac{w_1g_1g_2}{g_1 + D_1} - \dfrac{w'g_2g_3}{g_2 + D_2} \\ -ng_3 + \dfrac{w_3g_2g_3}{g_2 + D_3} \end{pmatrix} + \begin{pmatrix} d_{11} & d_{12} & d_{13} \\ d_{21} & d_{22} & d_{23} \\ d_{31} & d_{32} & d_{33} \end{pmatrix} \begin{pmatrix} y_1 + 2x_1 - x_3 \\ y_2 + 2x_1 \\ y_3 - 2x_1 + x_2 - x_3 \end{pmatrix},
\tag{11}
$$

where

$$
d_{11} = h_1 - l + 2mg_1 + \frac{Dwg_2}{(g_1 + D)^2},
$$

$$
d_{12} = h_2 + \frac{wg_1}{g_1 + D},
$$

$$
d_{13} = h_3,
$$

$$
d_{21} = h_4 - \frac{D_1w_1g_2}{(g_1 + D_1)^2},
$$

$$
d_{22} = h_5 + a_2 - \frac{w_1g_1}{g_1 + D_1} + \frac{D_2w'g_3}{(g_2 + D_2)^2},
$$

$$
d_{23} = h_6 + \frac{w'g_2}{g_2 + D_2},
$$

$$
d_{31} = h_7,
$$

$$
d_{32} = h_8 - \frac{D_3w_3g_3}{(g_2 + D_3)^2},
$$

$$
d_{33} = h_9 + n - \frac{w_3g_2}{g_2 + D_3}.
$$

Now, the errors for the coupled system are given as

$$
\begin{cases} e_1 = y_1 + 2x_1 - x_3, \\ e_2 = y_2 + 2x_1, \\ e_3 = y_3 - 2x_1 + x_2 - x_3. \end{cases}
\tag{12}
$$

We provide a simple algorithm to achieve a GS in Appendix A.

## 5. Numerical Results

In this section, we explore the possible occurrence of chaotic dynamics and GS for both the HP and UR models for small and moderate initial conditions. We used MATLAB 2019a in performing our simulations. We consider the following parameter set for the numerical experiments:

$$
\begin{cases} a_1 = 1.93, b_1 = 0.06, w = 1, D = 10, a_2 = 1, w_1 = 2, D_1 = 10, w_2 = 0.405, \\ D_2 = 10, c = 0.03, w_3 = 1, D_3 = 20, l = 1.75, m = 0.05, n = 0.1, w' = 0.6. \end{cases}
\tag{13}
$$

We show an example of chaos in the UR model and HP model using the parameters in Equation (13) with the initial conditions $[x_1(0), x_2(0), x_3(0)] = [10, 10, 10]$ and $[y_1(0), y_2(0), y_3(0)] = [10, 10, 10]$ as shown in Figures 1 and 2, respectively.

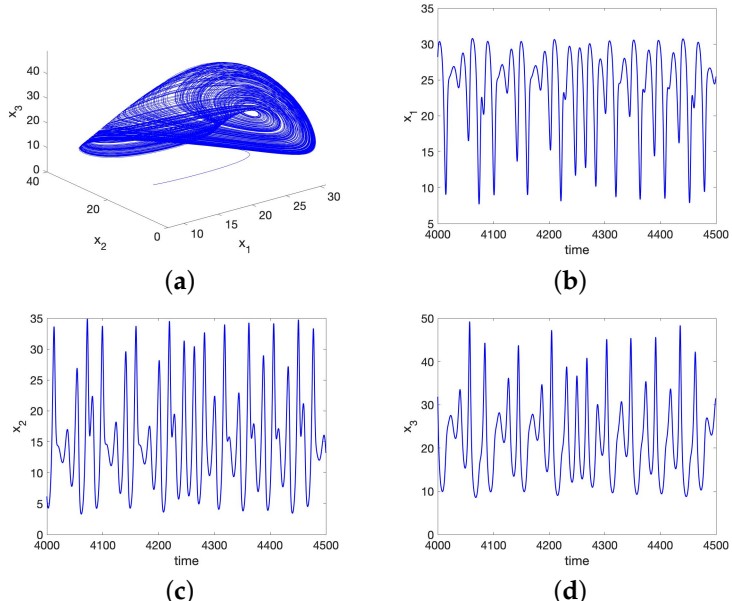

**Figure 1.** Chaotic dynamics with initial data $[x_1(0), x_2(0), x_3(0)] = [10, 10, 10]$ in UR model. (**a**) Phase diagram showing chaotic attractor in UR model. (**b**) Time series plot for $x_1$. (**c**) Time series plot for $x_2$. (**d**) Time series plot for $x_3$.

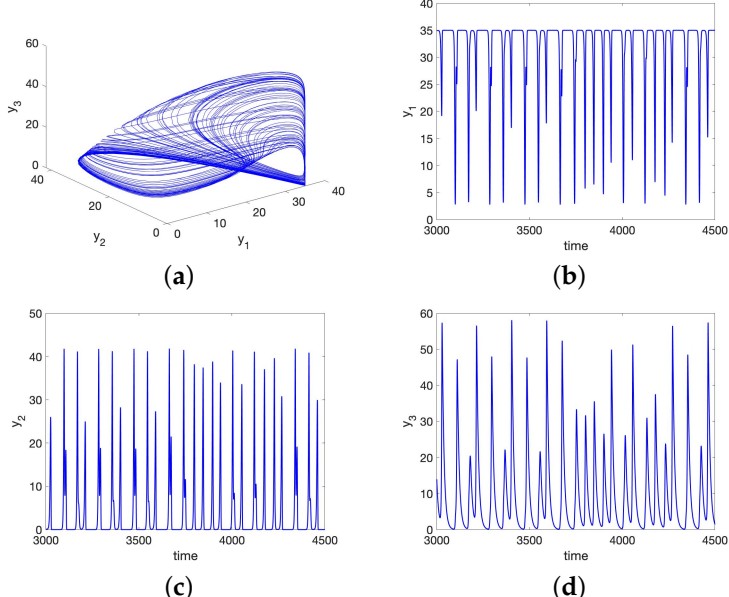

**Figure 2.** Chaotic dynamics with initial data $[y_1(0), y_2(0), y_3(0)] = [10, 10, 10]$ in HP model. (**a**) Phase diagram showing chaotic attractor in HP model. (**b**) Time series plot for $y_1$. (**c**) Time series plot for $y_2$. (**d**) Time series plot for $y_3$.

### 5.1. Chaos in the UR Model and HP Model for Small Initial Data

In this subsection, we investigate the likelihood of observing chaotic dynamics in both the UR and HP models. Let us consider the small initial data $[x_1(0), x_2(0), x_3(0)] = [0.01, 0.01, 0.1]$ and $[y_1(0), y_2(0), y_3(0)] = [0.01, 0.01, 0.01]$ for the UR and HP models, respectively. From Figures 3 and 4, our numerical simulations show the chaotic dynamics in the UR and HP models, respectively. The parameters used for the simulation can be seen in Equation (13).

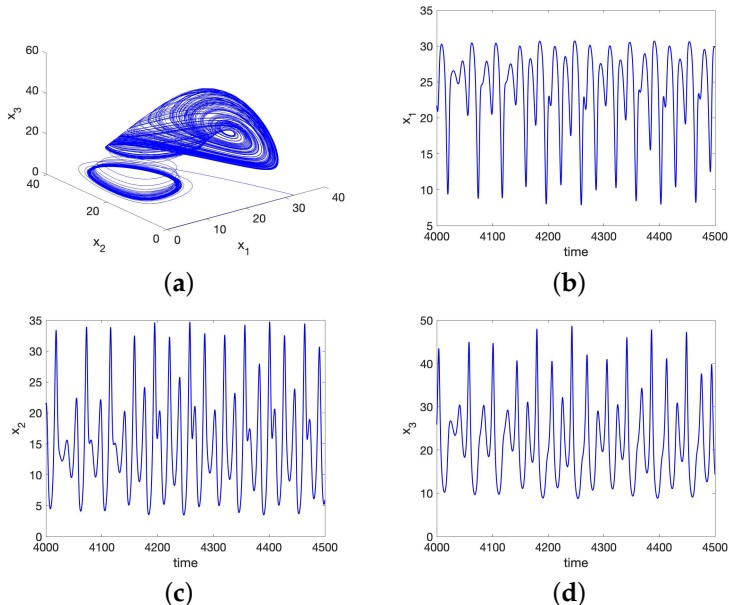

**Figure 3.** Chaotic dynamics with small initial data $[x_1(0), x_2(0), x_3(0)] = [0.01, 0.01, 0.1]$ in UR model. (**a**) Phase diagram showing chaotic attractor in UR model. (**b**) Time series plot for $x_1$. (**c**) Time series plot for $x_2$. (**d**) Time series plot for $x_3$.

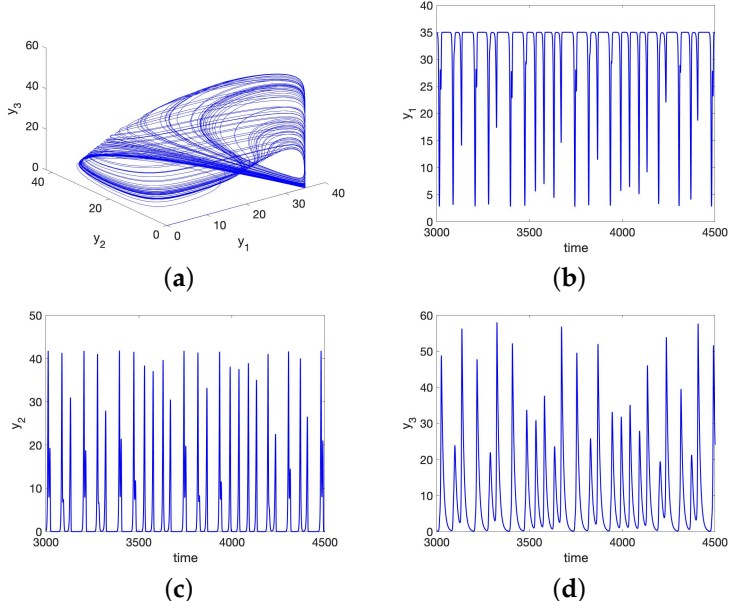

**Figure 4.** Chaotic dynamics with small initial data $[y_1(0), y_2(0), y_3(0)] = [0.01, 0.01, 0.01]$ in HP model. (**a**) Phase diagram showing chaotic attractor in HP model. (**b**) Time series plot for $y_1$. (**c**) Time series plot for $y_2$. (**d**) Time series plot for $y_3$.

### 5.2. GS for the UR Model and HP Model for Small Initial Data

We used the following small initial data for the GS of the systems in Equations (5) and (6), where $[x_1(0), x_2(0), x_3(0)] = [0.01, 0.01, 0.1]$ and $[y_1(0), y_2(0), y_3(0)] = [0.01, 0.01, 0.01]$, respectively. We observed a one-to-one correlation between the response variables and the goal dynamics, ensuring GS in Figure 5a–c. The parameters used are in Equation (13). Note that a one-to-one correlation is the period/frequency ratio between the response variables and their transformed driver states.

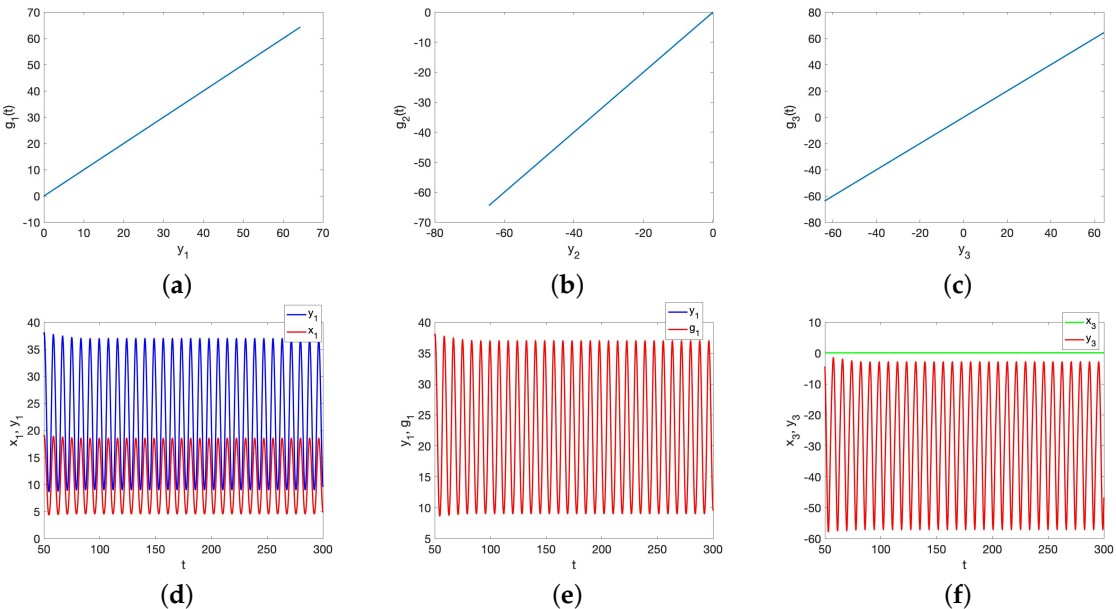

**Figure 5.** GS with small initial data. (**a**) $y_1 - g_1$ plot showing a 1:1 correlation. (**b**) $y_2 - g_2$ plot showing a 1:1 correlation. (**c**) $y_3 - g_3$ plot showing a 1:1 correlation. (**d**) Simulation showing time evolution for $x_1$ and $y_1$ after transients die out. (**e**) Simulation showing time evolution for $g_1$ and $y_1$ after transients die out. (**f**) Time series plot showing no blow-up in $x_3$ after transients die out.

## 6. Possible Causes of a Lack of Synchronization

We investigate the possible causes of no synchronization, as seen in Figure 6. Although the HP model has bounded solutions for any initial condition [20], this is not true in the case of the UR model. The UR model possesses the dynamics of a finite-time blow-up [24]. We present a few details for completeness.

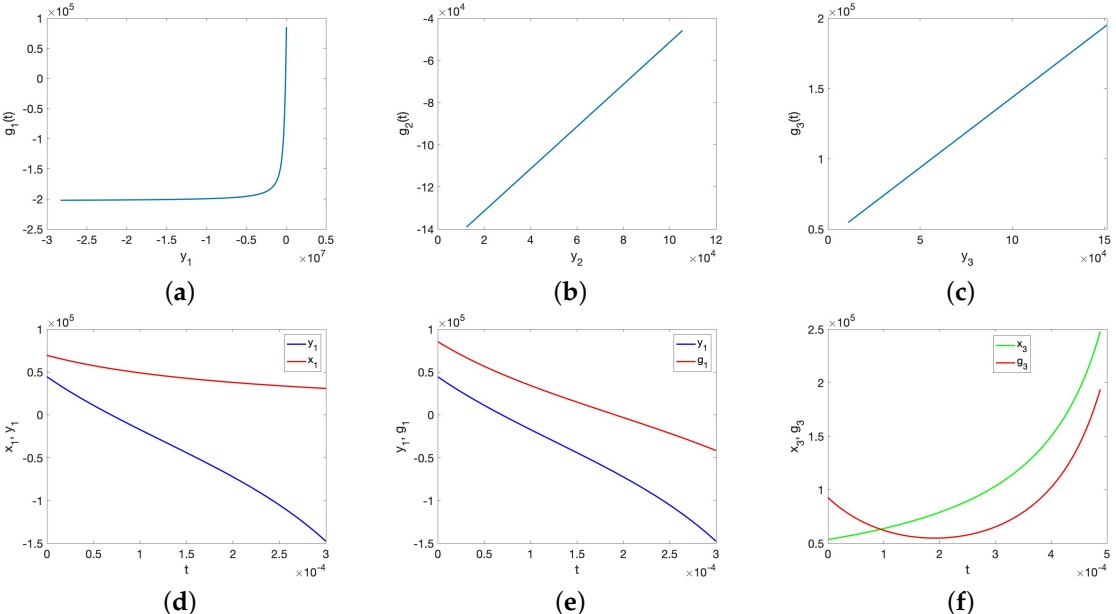

**Figure 6.** No GS with large initial data. (**a**) $y_1 - g_1$ plot showing no 1:1 correlation. (**b**) $y_2 - g_2$ plot showing no 1:1 correlation. (**c**) $y_3 - g_3$ plot showing no 1:1 correlation. (**d**) Simulation showing time evolution for $x_1$ and $y_1$. (**e**) Simulation showing time evolution for $g_1$ and $y_1$. (**f**) Time series plot showing blow-up occurring in $x_3$ and estimated at time $T^* \approx 0.00048$.

We first prove a variation of a classical result for this system:

**Theorem 1.** *The solution $x_3(t)$ to the top predator equation in Equation (5) blows up in a finite time such that*

$$\limsup_{t \to T^* < \infty} |x_3(t)| = \infty, \tag{14}$$

*for the initial data $(x_1(0), x_2(0), x_3(0))$, which is sufficiently large.*

**Remark 2.** *The proof follows [24].*

**Proof.** Consider Equation (5) with positive initial conditions.

By integrating the top predator equation for $x_3$, we obtain

$$-\frac{1}{x_3} + \frac{1}{x_3(0)} = -ct + w_3 \int_0^t \frac{ds}{x_1(s) + x_2(s) + D_3}$$

which gives

$$x_3 = \frac{1}{\frac{1}{x_3(0)} - ct + w_3 \int_0^t \frac{ds}{x_1(s) + x_2(s) + D_3}}.$$

We can then show the following function:

$$\psi(t) = \frac{1}{\frac{1}{x_3(0)} - ct + w_3 \int_0^t \frac{ds}{x_1(s) + x_2(s) + D_3}} \to 0 \text{ as } t \to T^* < \infty. \tag{15}$$

This will then show that the solution $x_3$ will blow up at the finite time $t = T^*$. Essentially, for a chosen $(x_1(0), x_2(0))$ that is sufficiently large, there exists a $\delta > 0$ such that

$$\frac{1}{\frac{1}{x_3(0)} - ct + w_3 \int_0^t \frac{ds}{x_1(s) + x_2(s) + D_3}}$$
$$< \frac{1}{x_3(0)} - \frac{c}{2}t, \quad \text{for all } t \in (0, \delta). \tag{16}$$

This is because under the continuity of the state variables $x_1, x_2$, we have the following for when $\delta$ is sufficiently small and $t \in (0, \delta)$:

$$w_3 \int_0^t \frac{ds}{x_1(s) + x_2(s) + D_3} < \frac{ct}{2} \tag{17}$$

and thus

$$w_3 \int_0^t \frac{ds}{x_1(s) + x_2(s) + D_3} < \frac{c}{2}. \tag{18}$$

If $x_3(0)$ is then chosen to be sufficiently large, then we can find $T^{**} \in (0, \delta)$ such that

$$\frac{1}{x_3(0)} - \frac{c}{2}T^* = 0 => \boxed{T^* = \frac{2}{cx_3(0)}}.$$

Now, through application of the classical intermediate value theorem on the continuous function $\psi$, we obtain the existence of some $T^* \in (0, \delta)$, $T^* < T^{**}$, s.t $\psi(T^*) = 0$. This implies that $x_3(t)$, the solution to the top predator equation (Equation (5)), blows up in a finite time at $t = T^*$, and the theorem is proven. $\square$

**Remark 3.** *Note that an explicit representation of the blow-up time $T^*$ is derived in terms of the initial conditions. Essentially, $T^* = \frac{2}{cx_3(0)}$, and thus the larger the initial condition for the top predator $x_3$ in Equation (5), the quicker the blow-up occurs. Additionally, through the "large" initial data in Theorem 1, we mean we can find a constant large C such that $||(x_1(0), x_2(0), x_3(0))||_\infty > C$.*

**Theorem 2.** *The models in Equations (5) and (6) cannot synchronize for arbitrarily large initial conditions, where Equation (5) is considered the driver system.*

**Proof.** We proceed by contradiction. Assume that Equations (5) and (6) do indeed synchronize. Then, a necessary and sufficient condition for the synchronization is that a Lyapunov function $V$ exists for the response system. A standard one considered in most works on synchronization is

$$V = (e_1)^2 + (e_2)^2 + (e_3)^2. \tag{19}$$

By definition of this Lyapunov function, we have

$$V \geq (e_1)^2 = (y_1 + 2x_1 - x_3)^2 > (x_3)^2 - (y_1 + 2x_1)^2. \tag{20}$$

Thus, we obtain

$$\limsup_{t \to T^* < \infty} |x_3(t)|^2 < \limsup_{t \to T^* < \infty} |V(t)|^2 + \limsup_{t \to T^* < \infty} |y_1 + 2x_1|^2 \tag{21}$$

where $T^*$ is the blow-up time, as derived in Theorem 1. Note that by a simple comparison, we have

$$|y_1 + 2x_1|^2 < 2\left(|y_1|^2 + |2x_1|^2\right) < C < \infty. \tag{22}$$

This follows from the standard estimates [20,24]. However, under Theorem 1, we have

$$\limsup_{t \to T^* < \infty} |x_3(t)| = \infty \implies \limsup_{t \to T^* < \infty} |x_3(t)|^2 = \infty. \tag{23}$$

Thus, we must have via Equation (21) that

$$\limsup_{t \to T^* < \infty} |V(t)|^2 = \infty, \tag{24}$$

and thus one cannot have $\frac{dV}{dt} < 0$, and $V$ cannot be a Lyapunov function, which is a contradiction. This proves the result. $\square$

*No Synchronization for the UR Model or HP Model for Large Initial Data*

We use the following large initial data to investigate the possible occurrence of GS between the systems in Equations (5) and (6). Consider $[x_1(0), x_2(0), x_3(0)] = [69516, 49912, 53580]$ and $[y_1(0), y_2(0), y_3(0)] = [44518, 12390, 49036]$. We do not see a one-to-one correlation between the response variables and the goal dynamics to ensure GS, as seen in Figure 6a–c. The parameters used are in Equation (13):

**Remark 4.** *The initial conditions for the HP model were selected from its basin of attraction. In the chaotic attractor case, this was all of $\mathbb{R}_+^3$. This was because the chaotic attractor was globally attracting. However, caution is required with the UR model due to the blow-up dynamic. The basin of attraction for the UR model is actually an open problem. Essentially, for the UR model, the large initial condition would blow-up and thus not be in the basin of attraction. Thus, we selected the initial conditions numerically; that is, we tested what initial conditions would blow up versus which ones would not numerically to ascertain what initial data should be used to show synchronization versus a lack thereof.*

Consider a modified UR model given by

$$
\begin{aligned}
\dot{x_1} &= a_1 x_1 - b_1 x_1^2 - \frac{w x_1 x_2}{x_1 + D}, \\
\dot{x_2} &= -a_2 x_2 + \frac{w_1 x_1 x_2}{x_1 + D_1} - \frac{w_2 x_2 x_3}{x_2 + D_2}, \\
\dot{x_3} &= c x_3 - \frac{w_3 x_3^2}{x_2 + D_3}.
\end{aligned}
\tag{25}
$$

Here, the top predator grows linearly as opposed to quadratically, yielding bounded solutions for all times [30]. We used the GS scheme in Equation (3) to couple the modified UR model in Equation (25) and the HP model in Equation (6).

For the numerical simulations, we used the parameter set in Equation (13) and obtained the following dynamics in Figure 7a–c.

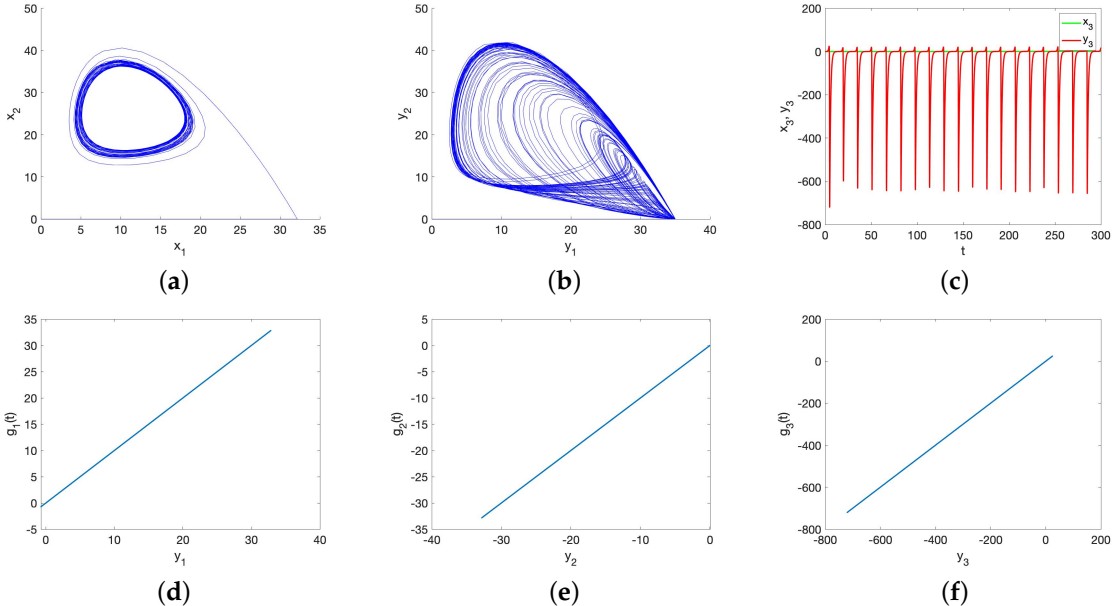

**Figure 7.** GS for modified UR model and HP model with initial data $[x_1(0), x_2(0), x_3(0)] = [0.0055, 0.0014, 0.0015]$ and $[y_1(0), y_2(0), y_3(0)] = [0.0026, 0.0084, 0.0025]$. (**a**) Phase plot for $x_1$ and $x_2$. (**b**) Phase plot for $y_1$ and $y_2$. (**c**) Time series plot for $x_3$ and $y_3$. (**d**) $y_1 - g_1$ plot showing 1:1 correlation. (**e**) $y_2 - g_2$ plot showing 1:1 correlation. (**f**) $y_3 - g_3$ plot showing 1:1 correlation.

Figure 7d–f shows GS in the systems in Equations (6) and (25) after coupling.
Now, let us consider another parameter set:

$$\begin{cases} a_1 = 1, b_1 = 0.6, w = 10, D = 0.1, a_2 = 1, w_1 = 20, D_1 = 0.1, w_2 = 4.05, \\ D_2 = 10, c = 0.03, w_3 = 1, D_3 = 20, l = 1.75, m = 0.05, n = 0.1, w' = 0.6. \end{cases} \tag{26}$$

From the numerical simulations shown in Figure 8d–f, when using the parameter set in Equation (26), there was GS in the modified UR model and HP model for small data.

We used the parameter set in Equation (13) to simulate the coupled modified UR model in Equation (25) and the HP model in Equation (6) for large initial data. The results are presented in Figure 9.

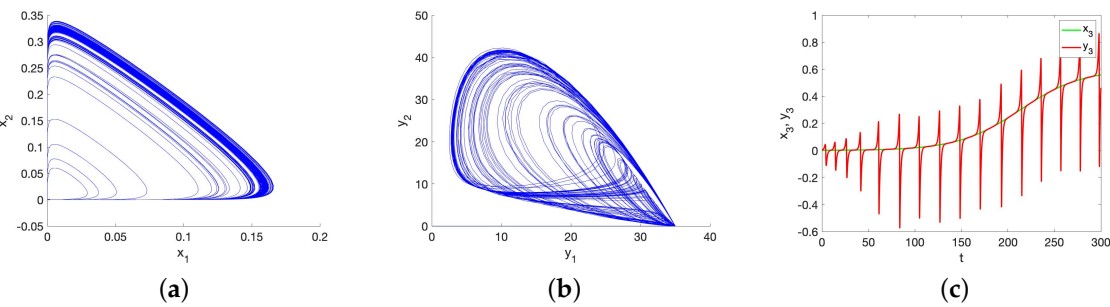

**Figure 8.** *Cont.*

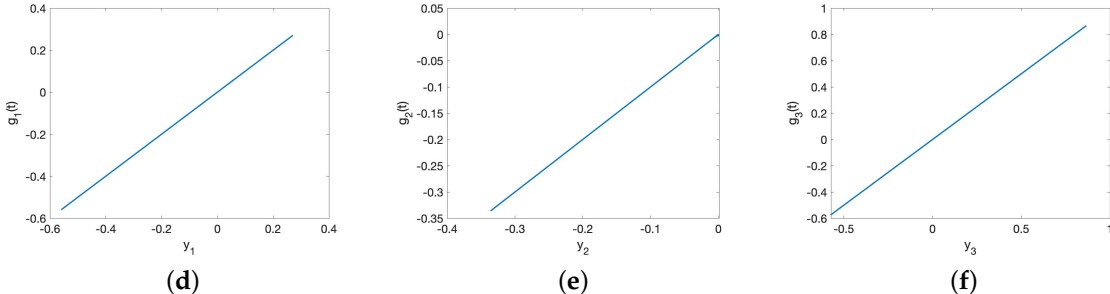

**Figure 8.** GS for modified UR model and HP model with initial data $[x_1(0), x_2(0), x_3(0)] = [0.001, 0.001, 0.001]$ and $[y_1(0), y_2(0), y_3(0)] = [0.001, 0.001, 0.001]$. (**a**) Phase plot for $x_1$ and $x_2$. (**b**) Phase plot for $y_1$ and $y_2$. (**c**) Time series plot for $x_3$ and $y_3$. (**d**) $y_1 - g_1$ plot showing 1:1 correlation. (**e**) $y_2 - g_2$ plot showing 1:1 correlation. (**f**) $y_3 - g_3$ plot showing 1:1 correlation.

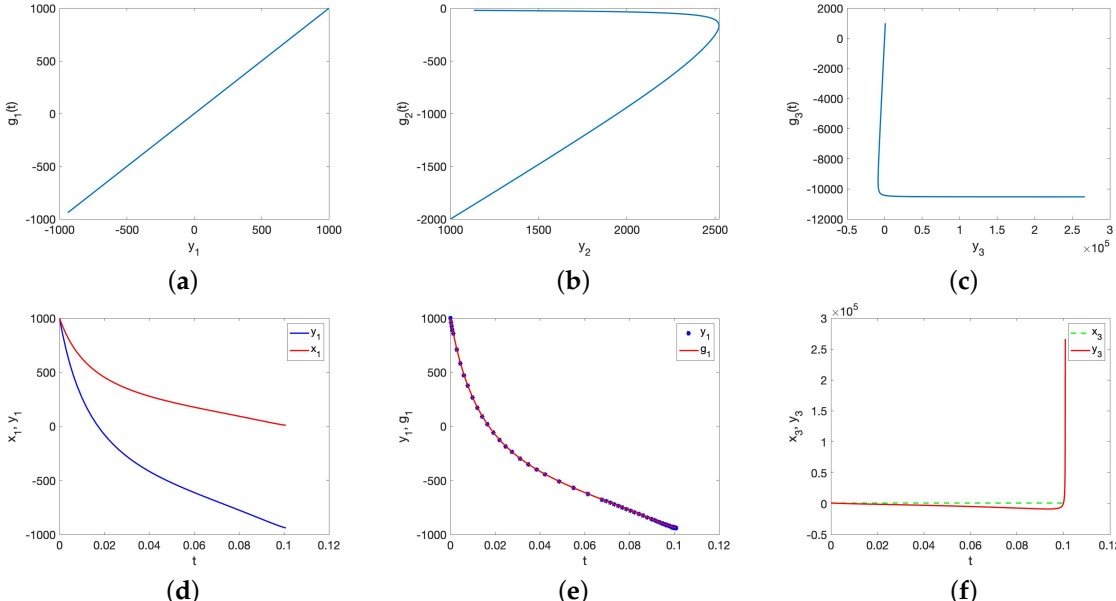

**Figure 9.** No synchronization for modified UR model and HP model with initial data $[x_1(0), x_2(0), x_3(0)] = [1000, 1000, 1000]$ and $[y_1(0), y_2(0), y_3(0)] = [1000, 1000, 1000]$. (**a**) $y_1 - g_1$ plot showing 1:1 correlation. (**b**) $y_2 - g_2$ plot showing no 1:1 correlation. (**c**) $y_3 - g_3$ plot showing no 1:1 correlation. (**d**) Simulation showing time evolution for $x_1 - y_1$. (**e**) Time series plot for $y_1 - g_1$. (**f**) Time series plot showing blow-up occurring in $y_3$ and estimated at time $T^* \approx 0.1008$.

## 7. Discussion and Conclusions

Real-world species are composed of multiple populations distributed over different geographical locations coupled by migration. Chaotic dynamics reduces the degree of synchrony among populations and thus reduces the chance of all of them being wiped out in the event of an exogenous stochastic perturbation [31]. In the current manuscript, we explored synchronization phenomena in two different but related model population systems. The existence of chaos in the linear version of the UR model suggests that ecological chaos has a larger role to play than previously thought. The effect of animal movements among different patches is replaced by prey–predator interactions. No synchronization of chaotic dynamics in the linear version of the UR model with the HP model for large initial populations affirms that generalist predators are major contributors toward the balance of natural food webs.

Synchronization of chaotic dynamics in UR and HP model systems is dependent on the initial data (i.e., initial population numbers). For small initial data, the UR and HP models synchronized as well as the modified UR and HP models after coupling. This is seen in

Figures 5 and 8. We saw no synchronization when the UR and HP models were coupled for the large initial data. Similarly, the modified UR and HP models did not synchronize for the large initial data after coupling. This reaffirms that the synchronization of three-species food chains with differently behaving top predators is caused solely by the initial numbers of the species, particularly the top predator.

The lack of synchronization results can be seen in Figures 6 and 9. Note that the modified UR model had bounded solutions for all times, and yet we could see a lack of synchrony in Figure 9. This reaffirms that a lack of synchronization is not solely a feature of the blow-up dynamic but of the top predator being a generalist in UR models versus a specialist in HP models. It would be interesting to test this synchronization or lack thereof in other variations of the UR and HP models. Several improvements have been made to the UR class of models. The spatially explicit and ODE UR model has been shown to have bounded solutions in certain regimes of parameters or initial data via non-monotonic functional responses [32], as well via refuge effects [33]. Cannibalism effects have also been considered herein [34], and testing the synchronization of such improved models with HP models will make for interesting future work. Another possible direction is to think of realization of the dynamics displayed by the current models via an actual hardware circuit. This has been achieved in the neuronal modeling literature [12,13]. Although obtaining ecological data is much more of a challenge than neuronal data obtained from, for example, EEG devices, it does beg the question of how such devices could be thought about in the ecological setting.

An absence of synchrony for large populations means that the delicate balance of nature is destroyed in the event of a blow up in population numbers. It will be interesting to explore different synchronization behaviors for different initial conditions in chaotic food chain models. As future endeavors, we could consider spatial UR and HP model systems to explore waves of synchrony or no synchronization, which would help us gain insight into the loss of biodiversity and eventual species extinction.

**Author Contributions:** Conceptualization, R.D.P. and E.M.T.; formal analysis, E.M.T. and R.D.P.; visualization, E.M.T.; writing—original draft preparation, E.M.T., R.D.P., R.K.U. and V.R.; writing—review and editing, E.M.T., R.D.P., R.K.U. and V.R. All authors have read and agreed to the published version of the manuscript.

**Funding:** This research received no external funding.

**Data Availability Statement:** Not applicable.

**Conflicts of Interest:** The authors declare no conflict of interest.

### Appendix A

We give a simple algorithm for achieving GS via the following steps:

**Step 1.** Identify the driver and response systems.
**Step 2.** Compute the Jacobian of the response system.
**Step 3.** Construct an H-matrix from the Jacobian and choose values for the H-matrix such that the Routh–Hurwitz criterion is satisfied.
**Step 4.** Construct the $\alpha$ transformation matrix which ensures the desired goal dynamics.
**Step 5.** Propose a coupling to achieve the GS state.

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
