# Peer review of "Blow-Up Dynamics and Synchronization in Tri-Trophic Food Chain Models"

_algorithms, doi:10.3390/a16040180_

Round 1

Author Response

please see attached letter

Reviewer 2 Report

The article deals with the synchronization of two dynamic systems (food chain models). There are several comments on the article’s content:

1. The functions g(t) and g(x) are alternately used in the article. This is incorrect, since these are different mathematical objects.

2. There is no explanation that alpha is a matrix. This can only be seen from an example.

3. The Jacobian, or rather the Jacobian matrix, is denoted in the article by J(g(t)) and dG/dg, although they are the same in meaning.

4. Definition 3.1 at the chosen place in the article does not make sense, since the coupled models with the controller D(.) has not yet been written.

5. In the theoretical part, a vector of errors is defined and a condition on its norm is indicated, and in the example, a vector of norms appears that does not correspond to the theoretical part.

6. T* appears in Theorem 5.1, but this value is not described. Some norm for a scalar is used. What is this norm? Similarly for Theorem 5.2. “Consider the three species food chain…” begins both theorems. Theorems don't need to be formulated like that.

7. It would be nice to present an algorithm, this is not superfluous for the Algorithms journal.

8. Excessive self-citation in the article under review: 15/39 ~ 38%.

Author Response

Please see attached letter.

Reviewer 3 Report

The paper is well written.  However, there are certain observations:

(a)  In Page2, nearly 16 citations have been referred/ given (without order).  It will be better if they are given in order as well as the improvements done in each paper, for a better clarity to the readers.

(b)  The conclusion section is to be improved to express precisely the improvements/generalization of UR/HP models.

(c)  Some minor English checks needed.

Author Response

Please see attached letter.

Round 2

Reviewer 1 Report

It seems that the authors have declared the reviewers' comments. The paper can be accepted in its present form.

Author Response

thank you for your comments

Reviewer 2 Report

Top of page 4: e(t) = y(t) - g(t), i.e. e_k(t) = y_k(t) - g_k(t), k = 1,...,n. Eq. (12): e_k = | ... |. These definitions of errors are different and there is no correspondence with the theoretical part of the paper.

Further, in the theorems, there are norms || ... ||_2, but what are they for? Under the norm sign are not vectors, but scalar values, i.e. we have absolute values | ... |. Formally, this is || ... ||_2 in the one-dimensional case, but this only confuses the presentation of the results.

Theorem 6.2: Models (5) and (6)

How correct is infinity – C in equation (24)?

Author Response

Thanks for your comments. Please see the attached letter.

Round 3

Reviewer 2 Report

The article has been corrected in accordance with reviewer's comments.
Last time I didn't pay attention to the used notation limsup|x(t)| --> oo,
but I suggest choosing one of the options: |x(t)| --> oo or limsup|x(t)| = oo.

Author Response

Dear referee,

Thank you for your insightful remarks that have helped us further improve the quality of our manuscript. We have made the lim sup changes as per your suggestion. We trust you will find everything in order. 

Sincerely,

Rana Parshad.